# Influence of the Passive Stabilization of the Trunk and Upper Limb on Selected Parameters of the Hand Motor Coordination, Grip Strength and Muscle Tension, in Post-Stroke Patients

**DOI:** 10.3390/jcm10112402

**Published:** 2021-05-29

**Authors:** Anna Olczak, Aleksandra Truszczyńska-Baszak

**Affiliations:** 1Rehabilitation Clinic, Military Institute of Medicine, 128 Szaserów Street, 04-141 Warsaw, Poland; 2Social Academy of Science, 11 Łucka Street, 00-842 Warsaw, Poland; 3Faculty of Rehabilitation, Józef Piłsudski University of Physical Education in Warsaw, 00-968 Warsaw, Poland; aleksandra.truszczynska@awf.edu.pl

**Keywords:** cerebral stroke, hand rehabilitation, core stability, muscle tone, motor coordination

## Abstract

Objective: Assessment of the influence of a stable trunk and the affected upper limb (dominant or non-dominant) on the parameters of the wrist and hand motor coordination, grip strength and muscle tension in patients in the subacute post-stroke stage compared to healthy subjects. Design: An observational study. Setting: Stroke Rehabilitation Department. Subjects: Thirty-four subjects after ischemic cerebral stroke and control group-32 subjects without neurological deficits, age and body mass/ height matched were included. Main measures: The tone of the multifidus, transverse abdominal and supraspinatus muscles were assessed by Luna EMG device. A HandTutor device were used to measure motor coordination parameters (e.g., range of movement, frequency of movement), and a manual dynamometer for measuring the strength of a hand grip. Subjects were examined in two positions: sitting without back support (non-stabilized) and lying with stabilization of the trunk and the upper limb. Results: Passive stabilization of the trunk and the upper extremity caused a significant improvement in motor coordination of the fingers (*p* ˂ 0.001) and the wrist (*p* < 0.001) in patients after stroke. Improved motor coordination of the upper extremity was associated with an increased tone of the supraspinatus muscle. Conclusions: Passive stabilization of the trunk and the upper limb improved the hand and wrist coordination in patients following a stroke. Placing patients in a supine position with the stability of the affected upper limb during rehabilitation exercises may help them to access latent movement patterns lost due to neurological impairment after a stroke.

## 1. Introduction

Up to 50% of patients report impaired upper limb and hand function following a stroke [1,2,3]. Unfortunately, functional restoration of the upper extremity requires a long-lasting physiotherapy and often fails to meet patients’ expectations [4,5]. Thus, it is important to develop new therapies to improve the motor function of the upper extremity in stroke patients [6,7,8,9,10,11].

The stabilization of the human body is the basis for maintaining balance and makes it possible to perform selective, coordinated movements with parts of the body [12,13,14,15,16,17]. There are several elements to the body’s stability: central stabilization, which concerns the proper tension and work of the deep muscles of the trunk, and a stable trunk [16,17,18]. According to Bobath’s concept, a stable trunk is a counterbalance to the movements of all limbs [19,20]. It is very important that the trunk is stable and mobile at the same time, because only then can any selective movement be made [15,19,21]. In addition to arm and hand motor function deficits, trunk movement coordination disorders have been observed in patients with stroke [22]. In particular, movement of the lower limb is associated with contraction of the abdominal muscles, which stabilize the spine [23]. Stabilization of the human body is essential for maintaining balance and allows coordinated movements of body parts, including the human radio-carpal joint and palm [24]. For example, a previous study showed that the position of the forearm was particularly important for achieving higher grip strength: better parameters were noted in the transverse position and the lateral plane compared to the plane consistent with the body axis and horizontal position [25]. Similarly, Okunribidi et al. found that stabilization of the forearm was particularly important for achieving higher grip strength in healthy subjects [25]. Stabilization was also key for the restoration of the correct movement pattern and for regaining function of the hand in patients with impaired motor coordination [26,27]. Similarly, passive stabilization of the trunk core may also assist in improving hand motor function following a stroke. Precise hand movements also rely on the proper functioning and position of the shoulder blade, and stroke patients often show a weakness of the shoulder (scapular) stabilizers [28,29]. The supraspinatus muscle of the shoulder plays an important role in the abduction, flexion and external rotation of the arm. Brunnstrom (1970) in her book recalls the Souque’s phenomenon, discovered by him in 1916 and consisting in that elevation of the affected arm frequently caused the paralyzed finger to extend [30]. A different study has shown that elbow joint control is dependent on shoulder abduction and the strength of shoulder abduction affects the elbow-flexion torque and, with that, the reaching range of motion (work area) in patients after stroke [31]. Nijland, et.al. in 2010 presented a paper on the basis of which they determined functional recovery of the hemiplegic arm at 6 months can be predicted early in a hospital stroke unit within 72 h after stroke onset, using two simple tests, finger extension and shoulder abduction [32]. Tanzarella et al. (2020), presented a non-invasive command study structure for draining up to 14 external and internal hand muscles. They extended previous research (limited to a few muscles) on a common synaptic signal in pools of motor neurons in a large number of muscles. The results showed that the component of the set of decoded spine trains of motor neurons was positively correlated with the force for all subjects and tasks. By grouping pools of motor neurons from external or internal muscles, a score correlated with strength was also obtained. These observations show low-dimensional control of motor neurons in many muscles, which can be used to extract control signals in nerve coupling, which can help in hand rehabilitation, e.g., control of the exoskeleton of the hand in patients after stroke, among others [33]. Modern devices used for therapy often use the principle of biofeedback. Biofeedback engages the brain to work, as a result of which the brain begins to learn, finding new possibilities to control, for example, muscles deprived of central control of movements [34,35]. The results of the study conducted by Delph et al. (2013) showed that by far the most popular type of device among those analyzed in the study was a device that used a pneumatic actuator to direct the flexion/extension of the finger [36]. To improve the function of the hand, transcranial magnetic stimulation with the use of non-invasive devices is also used. A randomized, double-blind study showed an increase in the physiological activity of the brain in the areas near the stroke [37,38]. Thanks to this, it is possible to improve many functions and motor activities, such as hand pressure and movements of the upper limbs as well walking speed [39]. Nevertheless, post-stroke patients have limitations in the movement of the distal upper limb. Especially in extending the wrist and fingers, due to the lack of, decreased or increased tension of a spastic nature.

So far, the influence of trunk and shoulder stabilization on the parameters of wrist and hand movements in patients after stroke has not been described.

The aim of the study was to evaluate the influence of the stable position of the trunk and upper limb on the parameters of hand and wrist movement coordination, grip strength and muscle tension in subacute post-stroke patients compared to healthy subjects. The aim of the study was also to evaluate the parameters of the affected upper limb depending on whether the examined limb was dominant or non-dominant.

## 2. Materials and Methods

### 2.1. Study Design

This is an observational study. Measurements made in two different positions of the trunk and affected upper limb. It has been checked how the intervention (effectiveness of stabilization) influence on a change in parameters. The passive and active range of motion, the maximum range of movement (ROM) and the frequency of movement at the wrist and fingers, as well as the grip force and the tension of the muscles deeply stabilizing the trunk and shoulder joint (dependent variables) were assessed among post-stroke patients in the sitting (non-stabilized) and supine (stabilized) position (independent variables). A group of healthy people was examined to assess whether neurological deficits in people after a stroke might affect the results of motor coordination and grip strength.

### 2.2. Ethics

The study was carried out in the Teaching Department of Rehabilitation of the Military Medical Institute (MMI) in Warsaw, Poland. It was approved by and carried out in accordance with the recommendations of the Ethical Committee of the Military Medical Institute (approval number 6/MMI/2020). Prior to inclusion, all subjects were informed about the purpose of the study. Written informed consent was obtained from all subjects in accordance with the tenets of the Declaration of Helsinki.

### 2.3. Subjects

In total 100 people were examined before including. 34 people (27 stroke patients and 7 healthy) were excluded because of their functional condition. Finally, 66 males and females were prospectively recruited from among patients of the Teaching Department and of the Physical Medicine Department of the MMI. Participation in this study was voluntary. The flow of participants through each stage of the study is shown below (Figure 1).

In this case, 34 subjects after ischemic cerebral stroke, (aged 44–83 years; mean, 64.1 ± 9.2 years); biometric data in Table 1. Study group were in the subacute (5–7 weeks past stroke) of the disease, with stable trunk (the Trunk Control Test 74–100 points), subjects were in a functional state allowing movements of the upper extremity (FMA-UE 43–49 motor function points); tension of forearm and hand muscles measured with Modified Ashworth Scale (MAS 1/1+) [40,41,42]. The clinical evaluation of patients after a stroke was performed by the physician admitting the patient to the clinic on the day of admission.

The control group comprised 32 subjects neurologically healthy patients (aged 35–74 years; mean, 58.6 ± 11.9 years); biometric data in Table 1. The control group was highly functional with stable trunk (the Trunk Control Test 100 points), (FMA-UE 66 motor function points); tension of forearm and hand muscles measured with Modified Ashworth Scale (MAS 0) [40,41,42].

The characteristics of the subjects are shown in Table 1.

Criteria for stroke group inclusion (1) patients with ischemic stroke; (2) patients with hemiparesis after 5 to 7 week after stroke; (3) subjects with stable trunk (the Trunk Control Test 70–100 points); (4) subjects who were in a functional state allowing movements of the upper extremity (FMA-UE 40–66 motor function points); (5) muscle tension (MAS 0–1+); (6) no severe deficits in communication, memory or understanding what can impede proper measurement performance; (7) at least 35 years of age; (8) maximum 83 years of age.

Criteria for stroke group exclusion: (1) stroke up to five weeks after the episode, (2) epilepsy, (3) lack of trunk stability, (4) no wrist and hand movement, (5) high or very low blood pressure, dizziness, malaise, (6) local infection of skin in the hand region.

Criteria for control group inclusion: (1) the control group consisted of subjects free from the upper extremity motor coordination disorders (FMA-UE 60–66 motor function points); subjects with stable trunk (TCT 100 points); correct muscle tension (MAS 0); (2) at least 35 years of age; (3) maximum 83 years of age.

Criteria for control group exclusion: (1) a history of neurologic or musculoskeletal disorders such as carpal tunnel syndrome, tendonitis, stroke, head injury or other conditions that could affect their ability to active movement and grip hand; (2) back pain; (3) no severe deficits in communication, memory or understanding what can impede proper measurement performance; (4) high or very low blood pressure, dizziness, malaise, (5) local infection of skin in the hand region.

### 2.4. Apparatus

The research was carried out according to the protocol no 3/KRN/2019, registered in Clinical Trial Registration.

A Luna EMG (a rehabilitation-diagnostic robot developed by EGZOTech, Gliwice, Poland) was used to measure muscle tension (accuracy of measurement [−1–+1 V+/−1 mV]). A manual electronic dynamometer (EH 101; Camry, Shiqi, China) was used for grip strength measurement (error of measurement 0.5 [kg/lb]). A Hand Tutor device composed of a safe and comfortable glove (with sensitive electro optical sensors evaluating position, speed wrist and finger movement; power supply: voltage: 5 [V] DC, rated current input: 300 [mA]), and the Medi Tutor TM software. A Hand Tutor was used to measure the kinematic parameters like: range of passive and active movement, deficits of movement (sensitivity: 0.05 [mm] of wrist and fingers Ext./Flex) as well as the speed/frequency of movement (motion capture speed: up to 1 [m/s]). The system (MediTouch, Israel) is used by many leading physical and occupational therapy centers worldwide and has CE and FDA certification [43]. The Hand Tutor glove was worn on the hand of the directly affected side in stroke patients and on the dominant extremity control group. Surface electrodes (single-use 55′ and ‘40 mm; EMG Electrodes; Sorimex, Poland) were affixed to the subject’s body according to the SENIAM (Surface ElectroMyoGraphy for the Non-Invasive Assessment of Muscles) procedure on the transverse abdominal, multifidus and supraspinal muscles (on the side directly affected in stroke patients). Before each exercise, the subject was instructed on how the exercise should be done.

### 2.5. Measurements

The examination consisted of two motor tasks, carried out in two different starting positions: sitting and lying down (supine). During the first examination, the subject sat on the therapeutic table (without back support), feet resting on the floor. The upper limb was to be examined in adduction of the humeral joint, with the elbow bent in the intermediate position between pronation and supination of the forearm. The wrist and the hand were free (no stabilization) but patients were asked to keep the wrist and hand in the extension of the forearm (as shown in Figure 2). 

The described position of the upper extremity is typically impossible to achieve in stroke patients without stabilization. Instead, features of flexion synergy (abduction in the humeral joint) were observed.

In the supine position, the upper limb was stabilized at the subject’s body (adduction in the humeral joint, elbow flexion in the intermediate position between pronation and supination of the forearm, with free wrist and the hand). In this position, patients were also asked to keep the wrist and hand in the extension of the forearm during each measurement (movement and grip strength) (Figure 3).

First, the range of passive in the radial-carpal joint (flexion and extension) and fingers (global flexion and extension) was measured in each position (sitting or supine) using the Hand Tutor device. The device also measured flexion/extension deficit refers to the difference between passive and active ROM.

Then the subject made active movements in the same order. Finally, the subject was asked to make moves as quickly and in as full a range as possible. The measurement of grip strength with a dynamometer was performed in both positions (sitting or supine) after the range of motion and speed/frequency tests. The reaction of the examined multifidus, transverse abdominal and supraspinatus muscles (tension values reported in microvolts [µV]) was also assessed during each of the exercise tasks using the surface electrodes (i.e., during movement of the wrist and during movement of the fingers).

### 2.6. Sample Size Calculation

The sample size was estimated using the G * Power 3.1.9.4 program. Assuming the following parameters: Effect size d = 0.9, α = 0.05; Power = 0.95 for the Wilcoxon-Mann-Whitney test, the required sample size was 58 (29 people per group).

### 2.7. Statistical Analysis

All statistical analyses were performed using the IBM SPSS Statistics suite, version 25 (IBM Corporation, Armonk, NY, USA). All data was analyzed using basic descriptive statistics. The distribution normality of the data was determined. The consistency with a normal distribution was verified with the Shapiro-Wilk test. Comparisons between two groups were performed using the *U* Mann-Whitney test. The Wilcoxon signed-rank test was used to compare measurements made in the non-stabilized and stabilized positions. A *p*-value of <0.05 was considered statistically significant.

## 3. Results

First, the effect of stabilization of the trunk and upper limb on the motor coordination of the hand and wrist was assessed in stroke patients (Table 2). Stabilization had no effect on the passive and active movement of the wrist and fingers or the deficits of extension and flexion. However, the frequency of movement of fingers 5, 4, 3 and 2 was higher in a stabilized position. Meanwhile, the maximum range of motion (ROM) of fingers 3 and 4 was higher in the non-stabilized position.

Next, the effect of stabilization of the trunk and upper limb on the movement of the hand and wrist in neurologically healthy subjects was assessed (Table 3). Active and passive movement of finger 5 and the wrist extension deficit were higher in the seated, non-stabilized position. Meanwhile, a higher maximum range of wrist movement was observed in the supine position with stabilization of the upper limb.

The effects of neurological deficits on the motor coordination of the hand and wrist were determined by comparing the stroke patients with a group of neurologically healthy subjects. Neurologically healthy subjects showed more significantly higher values (range of active wrist movement and frequency of wrist movements) in an unstable position. On the other hand, in a stabilized position, neurologically healthy subjects showed higher values only in the maximum range of wrist movement. Patients after stroke obtained much more results, important for the assessment of movement coordination, in a stable position. Even though the results show lower values than in healthy people, in the post-stroke group they are statistically significant and prove the advantage of a stable position of the trunk and upper limb for coordinated movement.

In stroke patients, the tone of the multifidus muscle was higher in the non-stabilized position than the stabilized position when the subject was moving their wrist or fingers (Table 4). Similarly, the tone of the multifidus, transverse abdominal and supraspinatus muscles was higher when moving the wrist or fingers in the non-stabilized position than the stabilized position in neurologically healthy subjects (Table 4). Neurologically healthy subjects had a higher level of muscle tone for all measured parameters than stroke patients in the non-stabilized position. However, in the stabilized position, neurologically healthy subjects exhibited a higher muscle tone for the multifidus and supraspinatus muscles during movement of the wrist or fingers than stroke patients.

In stroke group in whom the affected/dominant hand was examined, the range of flexion of fingers 3 and 4 was higher in the stabilized position (Table 5). Meanwhile, the range of the active movement of finger 4 was higher in the non-stabilized position. In subjects in whom an affected/non-dominant hand was examined, higher values were found for the frequency of movements from flexion to extension of fingers 2, 3, 4, 5, were observed in a stabilized position compared to a non-stabilized position. If the affected/dominant hand was tested, stroke group showed no significant differences in muscle tone depending on the position (i.e., seated or supine; Table 6). However, if the affected/non-dominant extremity was tested, higher muscle tones were found in a non-stabilized position than the stabilized position for the transverse muscle when moving the wrist and for the multifidus and transverse muscles when moving the fingers. The test result may be influenced by a small group of patients with a dominant (18) and non-dominant (16) hand.

## 4. Discussion

This study shows that coordination of the hand and wrist in stroke patients were better with passive stabilization of the trunk and upper limb. Stabilization of the trunk and upper limb turned out to be also important for improving the coordination of the affected/non-dominant hands in stroke patients. Motor coordination was assessed using the HandTutor^TM^ as well as at work Carmeli at al. although in their work the device, apart from the function of the test, was assessed in terms of the effects of therapy [43]. Commonly accepted scales and tests, Trunk Control Test, the Fugl-Meyer scale and the MAS assessment were used to analyze the functional conditions of the respondents [40,41,42,44].

In general, in the conditions of stabilization of the trunk and upper limb, a significant improvement in coordination, especially of the fingers, was noted with the HandTutor parameters in the group of people after stroke. Healthy people in a stable position achieved only a significantly greater range of maximum wrist motion. Patients after stroke obtained much more results, important for the assessment of movement coordination, in a stable position and although the results show lower values than in healthy people, the post-stroke are statistically significant and prove the advantage of a stable position for achieving coordinated movement of the distal part of the upper limb.

Passive stabilization of the trunk (or trunk restraint) could allow stroke patients to access “normal” movement patterns that may have been lost due to neurological impairment [27]. In the cited work the authors evaluate the shoulder and elbow joint during trunk restraint and report improvement in ranges of elbow and shoulder joint movement increased in both groups [27].

Overall, in all subjects, the tension of the deep muscles of the trunk and the supraspinatus muscle of the shoulder was higher in the seated, non-stabilized position than in the supine, stabilized positions. In addition, neurologically healthy subjects had a higher level of muscle tone than stroke patients while stroke, in most cases, presented higher results of passive movement which is probably before that the stroke patients have lower tension of muscles than healthy people [45]. Stroke patients have had also higher deficits of finger extension what is a consequence of the difference between the active and passive extension. Stroke patients usually cannot perform fully active fingers extension, because of weaker extensors than flexors [28]. Due to disturbed muscle activations, particularly of the extrinsic extensors, were significantly affected by postural changes of the interphalangeal, but not metacarpophalangeal, joints [26].

Recent studies show that in order to improve the functioning of the hand, one should strive to maintain and restore equal results in terms of grasping and squeezing strength of the dominant and non-dominant hand [46,47]. Moreover, researchers suggest that upper limb rehabilitation exercises after a stroke include the non-dominant and dominant arms of the dominant upper limb [48]. In this study, the effect of stabilization on the parameters of motor coordination in dominant and non-dominant limbs in stroke patients was assessed. However, only the affected upper limb was examined, the opposite limb was not tested, which is a limitation of research in this topic.

Correspondingly, this study showed the supraspinatus muscle demonstrated higher activity than the core stability muscles when the hand or wrist are moving. Greater activation of the supraspinatus muscle was observed in a stabilized position, and this corresponds to improved coordination and strength of the upper extremity.

Indeed, core or trunk stability has been shown to affect the motor coordination of the flexion of the hip and shoulder of the non-paretic side and trunk flexion and extension chronic stage stroke patients and healthy controls [49]. Garrison, B. and Wade, E. in 2015 presented studies on the ability of inertial sensors to record measures of limb coordination in non-disabled persons while performing ADL-inspired tasks. In this way, they assessed the limb coordination as measured by the features of the time and frequency domains with regard to the tasks of the upper limbs and assessed the relative sensitivity of these measures to different task types [6].

In this study, in the sitting and lying position, the position of the upper limb was the same (in adduction of the humeral joint, with the elbow bent in the intermediate position between pronation and supination of the forearm, the wrist and the hand were free-no stabilization). Furthermore, both in patients after a stroke and in healthy ones in the sitting position (without stabilization) the tension muscles the transverse abdominal and multifidus were higher than in the supine (stable) position. Only the supraspinatus in patients after a stroke showed, insignificant, though higher tension when the wrist was working in a stable position (lying down). It seems that the position of the body is important for improving the parameters of movement coordination, expressed primarily in the frequency of movement.

In this study, statistically higher results concerning motor coordination and grip force were obtained for the non-dominant hand under the conditions of stabilization of the trunk and upper limb. At the same time, comparing the muscle tension analyzed for the dominant and non-dominant limbs of stroke patients, both during wrist and finger movements, statistically higher tensions of the transversus abdominis and multifidus muscles were obtained in conditions without stabilization. In addition, the supraspinatus tension was higher in these conditions, but this result turned out to be statistically insignificant.

In addition, this study showed that passive stabilization of the upper limb (i.e., holding it close to the body) is also important for restoring the movement and function of the hand after a stroke. Indeed, reduced muscle tension often leads to a characteristic body position in stroke patients, typically involving a protractive position of the shoulders. Moreover, recent studies show a causal relationship between stroke and subacromial impingement syndrome, which leads to irritation of the supraspinatus muscle [30,31]. Therefore, in stroke patients, the muscle that plays an important role in stabilizing the upper extremity is usually in a highly unfavorable condition for its functional restoration. The results of this work clearly emphasize that the stable position of the trunk and the examined non-dominant limb improves the parameters of motor coordination with lower trunk muscle tension in this position. In post-stroke patients the supraspinatus tension was higher in the stable position, although not statistically significant, but this is what seems important in stroke patients. The generally lower muscle tension in people after a stroke, in the supine position for supraspinatus, behaves differently and is higher. It is as if in this position they were trying to compensate for the deficiencies by using stronger muscle groups (flexion synergy), abductors, that is, e.g., supraspinatus, for the distal movement of the upper limb. Kordelaar et al. in 2012 was to identify how pathological limb synergies between shoulder and elbow movements interact with compensatory trunk movements during a functional movement with the paretic upper limb after stroke [50]. The importance of limb movement in restoring fitness after a stroke has already been emphasized. Kwakkel et al. (2003), based on the Fugl-Meyer score for the flaccid arm, investigated that optimal prediction of arm performance at 6 months can be made within 4 weeks after the onset of stroke [2]. Nijland et al. (2021) reported that shoulders abduction was associated with the recovery of distal movements of the upper limb since the movement of the proximal joint favored the recovery of the distal ones. In addition, the authors observed that the preservation of shoulder abduction and finger extension movements reflected the level of integrity of the corticospinal tract [5,31,32]. Smania et al. in 2009 proved that the active finger extension scale is a strong early predictor of recovery of daily life autonomy in patients with stroke [51].

The presence of shoulder abduction as a determinant for upper limb function may reflect the intralimb neural coupling between proximal and distal segments in motor control [32]. The challenge for improving stroke recovery is understanding how to optimally engage and modify surviving neural networks to provide new response strategies that compensate for tissue loss following injury [52]. The basis of this improvement is the plasticity of the brain [53].

Overall, this study indicates that placing patients in the supine position with the upper limb (shoulder) stabilized beside their body during rehabilitation work may help them to access latent movement patterns that could have been lost due to stroke. However, further investigation is required to quantitatively assess the benefits of such passive stabilization with in a stroke rehabilitation program involving a large cohort of patients.

### 4.1. The Value of the Study

Our study found that passive stabilization of the shoulder in addition to the trunk during stroke rehabilitation is important to restore the correct movement pattern and regain function of the hand.

### 4.2. Limitations of the Study

The limitation of the study was the examination of only one of the muscles in the shoulder girdle, the supraspinatus. Certainly, examining more muscles in the shoulder girdle and upper limb would allow for more conclusions.

The examination may also be limited by examining patients in a specific functional state (e.g., muscles tension MAS 1/1+), however, in order to assess the coordination of movement and the grip strength, a functional state is needed that is characterized by the possibility of any movement in the examined wrist joint and fingers. However, it would be good to investigate functionally diverse groups of patients. The research could provide more evidence.

Only the affected upper limb was examined. It was determined whether or not it was the dominant limb, but the opposite upper limb was not examined. Examination of the opposite limb in conditions with and without stabilization would increase the value of the study and is planned in future research.

### 4.3. Clinical Messages

Passive stabilization of both the shoulder and the trunk can improve hand and wrist coordination in patients following a stroke.Placing patients in the supine position with the upper arm held beside their body during rehabilitation work may help them to access latent movement patterns lost due to stroke.

## 5. Conclusions

Passive stabilization of the trunk and upper extremity should be used in rehabilitation programs to restore coordination of movement in the distal part of upper limb.

## Figures and Tables

**Figure 1 jcm-10-02402-f001:**
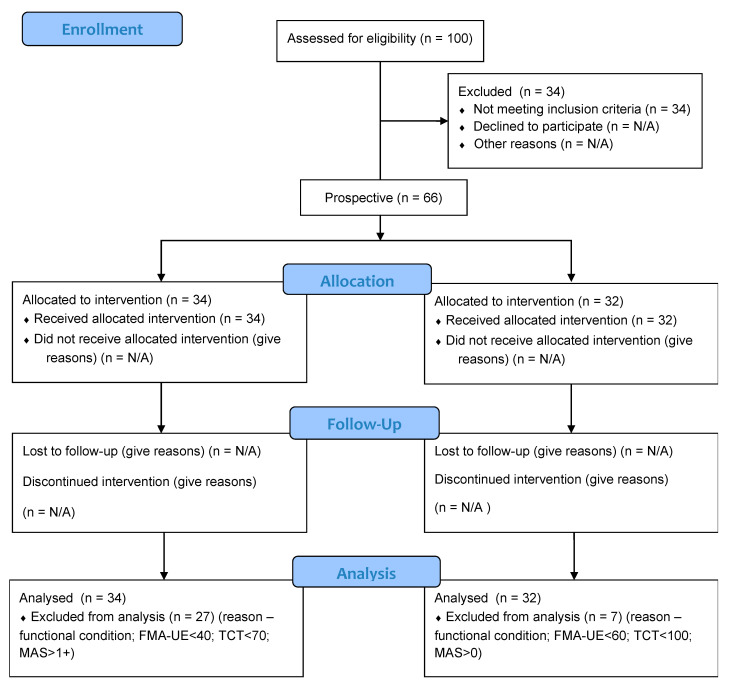
Flow of participants through each stage of the study.

**Figure 2 jcm-10-02402-f002:**
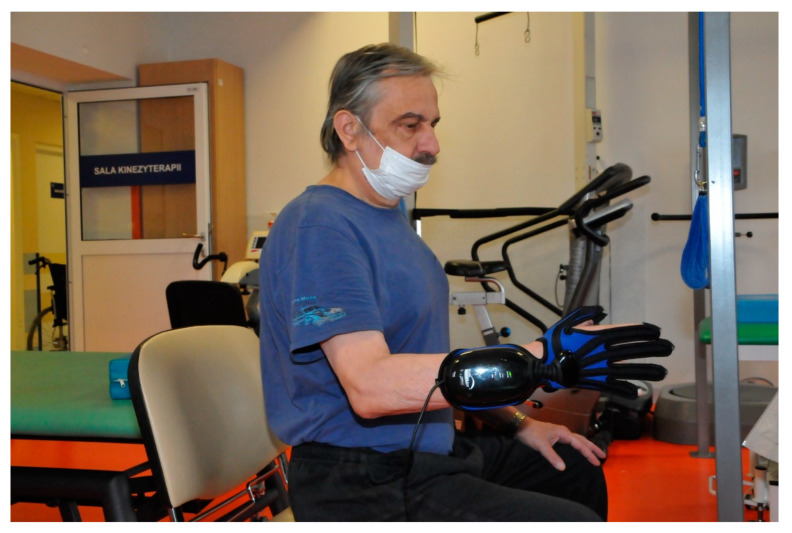
Sitting without back support (unstable position).

**Figure 3 jcm-10-02402-f003:**
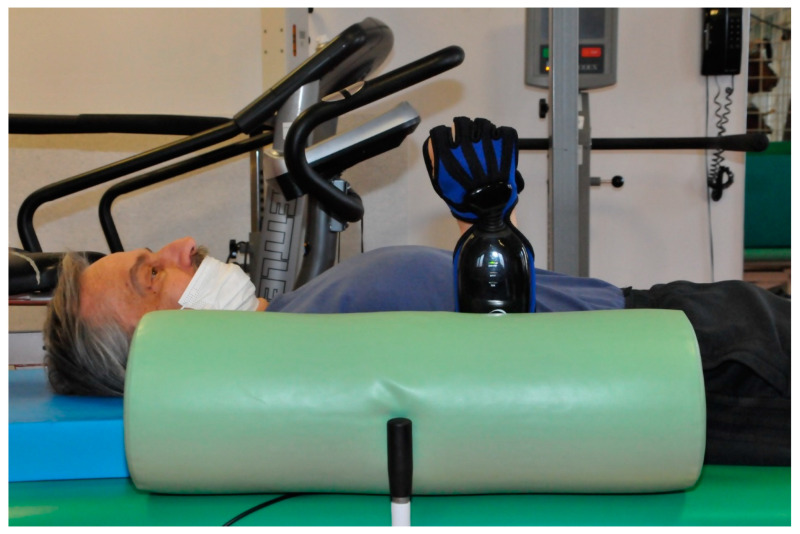
Passive stabilization of the trunk and upper limb (stable position).

**Table 1 jcm-10-02402-t001:** Biometric data of study population and clinical control group.

Group	Age	Height	Body Mass	BMI
Stroke	64.1 ± 9.2	167.7 ± 7.5	76.4 ± 9.4	27.16 ± 2.8
Control	58.6 ± 11.9	170.78 ± 8.8	72.50 ± 11.0	24.8 ± 2.7
Wilcoxon U	384.50	431.50	449.00	297.00
Z	−2.05	−1.45	−1.22	−3.17
*p*	0.040	0.148	0.222	0.002
effect size	0.25	0.18	0.15	0.39

**Table 2 jcm-10-02402-t002:** Motor coordination of the hand and wrist in stroke patients with and without stabilization of the trunk and upper limb.

	No Stabilization		Stabilization		*p* *
	M	SD	M	SD	
Range of active movement of the wrist, mm	17.91	6.17	18.74	6.28	0.374
Range of passive movement of the wrist, mm	25.35	3.76	25.38	4.48	0.922
5th finger active movement, mm	19.47	9.06	17.56	8.12	0.069
5th finger passive movement, mm	23.41	8.99	21.26	7.73	0.050
4th finger active movement, mm	23.03	7.38	21.56	7.80	0.062
4th finger passive movement, mm	27.62	6.72	25.35	6.79	0.069
3rd finger active movement, mm	22.94	6.05	22.47	5.85	0.253
3rd finger passive movement, mm	27.12	5.61	26.15	5.54	0.536
2nd finger active movement, mm	20.65	5.88	20.29	5.75	0.528
2nd finger passive movement, mm	27.00	5.50	25.09	4.44	0.085
1st finger active movement, mm	9.18	5.61	8.44	4.86	0.573
1st finger passive movement, mm	12.68	5.81	12.97	5.73	0.893
Wrist extension deficit, mm	4.21	3.72	3.76	3.55	0.250
Wrist flexion deficit, mm	3.24	2.68	2.88	3.07	0.281
5th finger extension deficit, mm	3.03	2.68	2.88	2.87	0.521
5th finger flexion deficit, mm	0.91	1.99	0.82	1.57	0.932
4thfinger extension deficit, mm	2.82	2.81	2.06	1.84	0.205
4th finger flexion deficit, mm	1.76	3.90	1.74	2.43	0.483
3rd finger extension deficit, mm	2.59	3.33	1.82	1.95	0.247
3rd finger flexion deficit, mm	1.59	3.15	1.85	2.45	0.433
2nd finger extension deficit, mm	3.65	4.46	1.94	1.91	0.061
2nd finger flexion deficit, mm	2.71	3.47	2.85	2.89	0.844
1st finger extension deficit, mm	1.26	1.69	1.21	1.18	0.889
1st finger flexion deficit, mm	2.24	3.47	3.32	4.00	0.239
Frequency of wrist movement (flexion to extension), cycles ^#^/s	1.15	0.71	1.21	0.86	0.698
Wrist maximum ROM, mm	19.73	10.73	19.94	9.36	0.153
Frequency of 5th finger movement (flexion to extension), cycles ^#^/s	1.55	0.96	1.67	0.94	0.016
5th finger maximum ROM, mm	18.05	8.50	17.39	10.78	0.124
Frequency of 4th finger movement (flexion to extension), cycles ^#^/s	1.55	0.96	1.69	0.96	0.011
4th finger maximum ROM, mm	22.70	8.04	20.56	7.74	0.018
Frequency of 3rd finger movement (from flexion to extension), cycles ^#^/s	1.53	0.98	1.68	0.96	0.007
3rd finger maximum ROM, mm	22.00	5.82	20.41	6.02	0.027
Frequency of 2nd finger movement (flexion to extension), cycles ^#^/s	1.54	0.98	1.69	0.96	0.008
2nd finger maximum ROM, mm	18.64	6.01	18.01	5.94	0.478
Frequency of the 1st finger movement (flexion to extension), cycles ^#^/s	1.17	0.95	1.19	1.04	0.453
1st finger maximum ROM, mm	9.30	4.78	8.52	4.24	0.745
Grip strength, kg	18.32	14.26	19.13	14.22	0.086

Legend: M—mean; ROM—range of motion; SD—standard deviation; * Wilcoxon test; ^#^ one cycle = the movement from flexion to contraction. Notes: Passive and active ROM is a sum of all the finger flexion or extension angles (i.e., at the MCP, PIP and DIP joints); the extension deficit refers to the difference between passive and active ROM.

**Table 3 jcm-10-02402-t003:** Motor coordination of the hand and wrist in neurologically healthy subjects with and without stabilization of the trunk and upper limb.

	No Stabilization		Stabilization		*p* *
	M	SD	M	SD	
Range of active movement of the wrist, mm	21.78	4.76	22.34	4.92	0.232
Range of passive movement of the wrist, mm	26.63	4.02	26.16	4.07	0.304
5th finger active movement, mm	18.75	5.86	17.50	5.86	0.011
5th finger passive movement, mm	20.09	6.54	18.84	6.44	0.005
4th finger active movement, mm	21.28	6.07	20.28	4.95	0.144
4th finger passive movement, mm	22.50	6.40	22.16	5.74	0.430
3rd finger active movement, mm	21.34	5.76	20.84	5.04	0.390
3rd finger passive movement, mm	22.72	6.22	22.84	5.58	0.654
2nd finger active movement, mm	20.16	4.13	19.88	3.37	0.704
2nd finger passive movement, mm	21.69	4.69	22.22	4.43	0.218
1st finger active movement, mm	7.06	3.54	7.09	2.88	0.339
1st finger passive movement, mm	9.31	4.13	9.16	3.92	0.524
Wrist extension deficit, mm	2.69	2.33	2.19	2.24	0.012
Wrist flexion deficit, mm	2.16	2.25	1.63	1.83	0.064
5th finger extension deficit, mm	1.00	2.13	0.81	1.42	0.495
5th finger flexion deficit, mm	0.34	0.90	0.53	1.32	0.523
4thfinger extension deficit, mm	0.75	1.39	1.06	1.52	0.134
4th finger flexion deficit, mm	0.47	0.98	0.81	1.93	0.518
3rd finger extension deficit, mm	0.72	1.61	0.94	1.37	0.256
3rd finger flexion deficit, mm	0.66	1.15	1.06	2.03	0.296
2nd finger extension deficit, mm	0.97	1.58	1.50	2.49	0.145
2nd finger flexion deficit, mm	0.56	0.84	0.84	1.67	0.558
1st finger extension deficit, mm	0.72	1.46	0.59	1.01	0.726
1st finger flexion deficit, mm	1.53	2.42	1.47	3.25	0.099
Frequency of wrist movement (flexion to extension), cycles ^#^/s	2.87	0.98	2.97	0.79	0.293
Wrist maximum ROM, mm	23.64	7.27	26.38	8.35	0.003
Frequency of 5th finger movement (flexion to extension), cycles ^#^/s	2.86	0.88	2.83	0.91	0.146
5th finger maximum ROM, mm	20.41	7.88	19.44	7.81	0.362
Frequency of 4th finger movement (flexion to extension), cycles ^#^/s	3.02	0.77	3.17	0.55	0.091
4th finger maximum ROM, mm	23.32	9.58	22.11	10.71	0.364
Frequency of 3rd finger movement (from flexion to extension), cycles ^#^/s	3.02	0.77	3.06	0.74	0.279
3rd finger maximum ROM, mm	21.16	5.62	22.27	7.34	0.337
Frequency of 2nd finger movement (flexion to extension), cycles ^#^/s	3.01	0.77	3.08	0.68	0.159
2nd finger maximum ROM, mm	19.00	4.23	19.52	5.20	0.896
Frequency of the 1st finger movement (flexion to extension), cycles ^#^/s	2.49	1.14	2.43	1.14	0.741
1st finger maximum ROM, mm	13.78	6.05	15.60	6.85	0.084
Grip strength, kg	35.25	11.88	34.33	11.99	0.286

Legend: M—mean; ROM—range of motion; SD—standard deviation; * Wilcoxon test; ^#^ one cycle = the movement from flexion to contraction. Notes: Passive and active ROM is a sum of all the finger flexion or extension angles (i.e., at the MCP, PIP and DIP joints); the extension deficit refers to the difference between passive and active ROM.

**Table 4 jcm-10-02402-t004:** Muscle tension parameters measured during wrist or finger movement, with and without stabilization of the trunk and upper limb.

	Stroke Group (*n* = 34)					Neurologically Healthy Group (*n* = 32)				
	No Stabilization		Stabilization		*p* *	No Stabilization		Stabilization		*p* *
	M	SD	M	SD		M	SD	M	SD	
Tension measured during wrist movement										
Multifidus muscle tension, µV	76.84	100.28	42.36	34.15	0.028	130.73	77.72	75.80	79.77	<0.001
Transverse muscle tension, µV	69.52	80.54	58.60	73.99	0.099	141.40	165.86	65.54	70.46	<0.001
Supraspinatus muscle tension, µV	81.15	73.16	83.88	61.64	0.278	170.19	95.92	120.78	47.91	<0.001
Tension measured during finger movement										
Multifidus muscle tension, µV	78.30	100.43	42.74	32.69	0.014	126.77	80.21	70.43	63.37	<0.001
Transverse muscle tension, µV	63.15	68.06	51.57	57.63	0.126	137.38	166.16	62.54	66.74	<0.001
Supraspinatus muscle tension, µV	82.29	72.52	80.34	61.04	0.215	165.55	95.03	115.84	52.19	<0.001

Legend: M—mean; SD—standard deviation; * Wilcoxon test.

**Table 5 jcm-10-02402-t005:** Effect of hand dominance on motor coordination of the hand and wrist in stroke patients with and without stabilization of the trunk and upper limb.

	Affected/Dominant Hand					Affected/Non-Dominant				
	No Stabilization		Stabilization		*p* *	No Stabilization		Stabilization		*p* *
	M	SD	M	SD		M	SD	M	SD	
Range of active movement of the wrist, mm	16.94	5.74	17.17	6.14	0.755	19.00	6.62	20.50	6.14	0.342
Range of passive movement of the wrist, mm	24.78	3.74	24.17	4.82	0.660	26.00	3.80	26.75	3.75	0.549
5th finger active movement, mm	16.17	7.23	15.56	8.10	0.552	23.19	9.67	19.81	7.77	0.083
5th finger passive movement, mm	19.17	6.71	19.56	7.60	0.856	28.19	8.98	23.19	7.64	0.017
4th finger active movement, mm	21.89	7.32	19.67	7.90	0.021	24.31	7.46	23.69	7.36	0.931
4th finger passive movement, mm	24.61	6.42	23.61	6.71	0.261	31.00	5.45	27.31	6.52	0.154
3rd finger active movement, mm	22.61	5.65	21.56	5.64	0.120	23.31	6.63	23.50	6.09	0.856
3rd finger passive movement, mm	24.67	5.36	25.44	5.19	0.324	29.88	4.63	26.94	5.97	0.111
2nd finger active movement, mm	19.28	5.03	18.50	5.69	0.377	22.19	6.53	22.31	5.28	0.888
2nd finger passive movement, mm	25.17	6.39	23.33	4.17	0.436	29.06	3.42	27.06	3.97	0.086
1st finger active movement, mm	8.89	5.08	8.06	4.11	0.494	9.50	6.31	8.88	5.70	0.875
1st finger passive movement, mm	12.11	5.87	12.56	5.45	0.867	13.31	5.86	13.44	6.19	0.979
Wrist extension deficit, mm	4.56	3.37	4.28	3.36	0.393	3.81	4.15	3.19	3.78	0.503
Wrist flexion deficit, mm	3.28	2.08	2.72	3.43	0.107	3.19	3.29	3.06	2.72	0.840
5th finger extension deficit, mm	2.56	2.28	3.17	2.94	0.728	3.56	3.05	2.56	2.85	0.275
5th finger flexion deficit, mm	0.44	1.20	0.83	1.62	0.394	1.44	2.56	0.81	1.56	0.438
4thfinger extension deficit, mm	2.33	2.54	2.11	2.08	0.887	3.38	3.07	2.00	1.59	0.091
4th finger flexion deficit, mm	0.39	0.70	1.83	2.66	0.028	3.31	5.30	1.63	2.22	0.349
3rd finger extension deficit, mm	1.78	2.05	1.78	2.24	0.681	3.50	4.24	1.88	1.63	0.180
3rd finger flexion deficit, mm	0.28	0.58	2.11	2.91	0.003	3.06	4.14	1.56	1.86	0.254
2nd finger extension deficit, mm	4.00	5.22	1.78	2.05	0.112	3.25	3.53	2.13	1.78	0.284
2nd finger flexion deficit, mm	1.89	1.97	3.06	2.94	0.234	3.63	4.52	2.63	2.92	0.309
1st finger extension deficit, mm	0.94	0.94	1.06	1.00	0.719	1.63	2.25	1.38	1.36	0.632
1st finger flexion deficit, mm	2.28	3.83	3.44	3.68	0.452	2.19	3.15	3.19	4.45	0.476
Frequency of wrist movement (flexion to extension), cycles ^#^/s	1.04	0.69	1.08	0.74	0.959	1.28	0.74	1.35	0.98	0.581
Wrist maximum ROM, mm	20.44	13.82	20.69	11.73	0.039	18.92	5.97	19.10	5.96	0.897
Frequency of 5th finger movement (flexion to extension), cycles ^#^/s	1.44	0.95	1.53	0.87	0.162	1.67	1.00	1.83	1.02	0.036
5th finger maximum ROM, mm	16.67	8.20	15.09	9.04	0.184	19.59	8.82	19.97	12.24	0.438
Frequency of 4th finger movement (flexion to extension), cycles ^#^/s	1.46	0.94	1.56	0.90	0.162	1.66	0.99	1.83	1.03	0.029
4th finger maximum ROM, mm	21.31	6.52	19.46	8.60	0.088	24.26	9.45	21.80	6.69	0.121
Frequency of 3rd finger movement (from flexion to extension), cycles ^#^/s	1.46	0.94	1.55	0.89	0.162	1.62	1.04	1.83	1.03	0.018
3rd finger maximum ROM, mm	21.83	5.30	19.87	6.25	0.149	22.18	6.53	21.03	5.89	0.173
Frequency of 2nd finger movement (flexion to extension), cycles ^#^/s	1.46	0.94	1.56	0.90	0.162	1.63	1.05	1.83	1.03	0.018
2nd finger maximum ROM, mm	18.04	5.74	17.36	6.58	0.632	19.31	6.41	18.74	5.24	0.552
Frequency of the 1st finger movement (flexion to extension), cycles ^#^/s	1.17	0.89	1.21	1.08	0.585	1.18	1.04	1.18	1.03	0.622
1st finger maximum ROM, mm	8.51	4.80	8.12	3.92	0.744	10.19	4.76	8.98	4.66	0.255
Grip strength, kg	15.54	12.28	16.27	12.74	0.372	21.44	16.03	22.34	15.50	0.088

Legend: M—mean; ROM—range of motion; SD—standard deviation; * Wilcoxon test; ^#^ one cycle = the movement from flexion to contraction. Notes: Passive and active ROM is a sum of all the finger flexion or extension angles (i.e., at the MCP, PIP and DIP joints); the extension deficit refers to the difference between passive and active ROM.

**Table 6 jcm-10-02402-t006:** Effect of hand dominance on muscle tension in stroke patients with and without stabilization of the trunk and upper limb.

Muscle Tension	Affected/Dominant Hand					Affected/Non-Dominant Hand				
	No Stabilization		Stabilization		*p* *	No Stabilization		Stabilization		*p* *
	M	SD	M	SD		M	SD	M	SD	
Tension measured during wrist movement										
Multifidus muscle tension, µV	63.47	80.52	42.85	35.12	0.199	91.88	119.66	41.82	34.17	0.063
Transverse muscle tension, µV	57.37	54.30	70.43	94.18	0.679	83.19	102.73	45.29	40.43	0.044
Supraspinatus muscle tension, µV	69.81	60.64	81.34	63.93	0.647	93.92	85.32	86.74	60.92	0.352
Tension measured during finger movement										
Multifidus muscle tension, µV	62.16	80.32	43.69	33.89	0.199	96.45	119.23	41.66	32.37	0.030
Transverse muscle tension, µV	51.47	52.89	59.84	71.18	0.983	76.30	81.69	42.27	37.28	0.026
Supraspinatus muscle tension, µV	71.75	61.09	75.16	64.24	0.396	94.14	84.01	86.17	58.74	0.326

Legend: M—mean; SD—standard deviation; * Wilcoxon test.

## Data Availability

Data available on request from corresponding author.

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
