# Peer review of "Influence of the Passive Stabilization of the Trunk and Upper Limb on Selected Parameters of the Hand Motor Coordination, Grip Strength and Muscle Tension, in Post-Stroke Patients"

_jcm, 2021, doi:10.3390/jcm10112402_

Round 1

Reviewer 1 Report

Line 44: change ‘stroke patients’ for ‘patients with stroke’.

Line 72: add a  ‘,’ after trunk and before and.

Subjects section: Why the authors choose the Wilcoxon-Mann-Whitney test for the calculation of the sample size? Are the authors assuming that you will obtain non-parametric data? Please, justify this in order to better understand the sample size specified in your study.

Line 126: The authors should specify in the inclusion criteria section that trunk stabilization and FMA-UE 43-49 motor function points are required to participate in this study.

137: I would like to suggest the authors to add a section entitled: ‘Apparatus’, before the measurements section.

Line 138: Is this protocol (protocol no 3/KRN/2019) registered in any source? If yes, please specify.

Line 284: change can’t for cannot.

Line 284-285: What about muscle spasticity after stroke? Most of time patients with stroke have limitations in performing fingers extension because of the spasticity. Did you measure spasticity levels of the hand, and forearm before to start the study? I think that this is an important characteristic to take into consideration regarding improvements in muscle tone and joint mobility. If the authors did not measure spasticity, this should be added as a limitations of the study section.  

Line 292-293: In addition to the dominant or non-dominant hand, the authors should specify the affected or non-affected side. Did the authors take this into consideration when assessing the patients? I mean, did you consider if the affected upper limb of the patients with stroke were their dominant or non-dominant side? How this could affect motor improvements of the upper limb? If not, this should be added in the limitations section too.

Line 297: Correlated? The authors did not use any correlation test in the statistical analyses.

Line 358: eliminate the or their.

Author Response

Manuscript ID: jcm-1173910

Title (previous): Hand function evaluation by HandTutor device and a manual dynamometer in stroke patients in supported and unsupported position

Full title (after change): Influence of a stable and unstable position of the trunk and upper limb on selected parameters of wrist and hand motor coordination, grip strength and muscle tension, in post-stroke and healthy subjects.

Dear Reviewers,

   Thank you very much for the analysis of our manuscript. We really appreciate your comments and indication of fragments that should be corrected and explained. Considering your suggestions, all mistakes were corrected. The introduction of corrections and changes in the text caused the numbering of the lines to shift. In response to reviewers' comments, it provides the original numbering. In order to avoid misunderstandings, changes introduced in the text are marked in blue and additionally, the manuscript was sent in the change tracking mode.

Reviewer #1:

Thank you very much for the very quick and thorough analysis of our manuscript.

The following comments and answers:

Line 44: change ‘stroke patients’ for ‘patients with stroke’.

Thank you very much. This suggestion was corrected.

Line 72: add a  ‘,’ after trunk and before and.

Thank you. The punctuation error has been corrected.

In response to the comment on the subjects section:

The sample size was calculated after the test was performed. A non-parametric test was used in the analyzes due to the skewed results of the measurements. On this basis, the sample size for which the power would be at a satisfactory level was estimated.

Thank you very much.

Line 126: The authors should specify in the inclusion criteria section that trunk stabilization and FMA-UE 43-49 motor function points are required to participate in this study.

Thank you very much for this review. The inclusion criteria have been corrected, supplemented, and moved to the Methods section.

Line 137: I would like to suggest the authors add a section entitled: ‘Apparatus’, before the measurements section.

Thank you very much for this suggestion. The new section called "Apparatus" has also been improved and completed.

Line 138: Is this protocol (protocol no 3/KRN/2019) registered in any source? If yes, please specify.

Protocol no 3 / KRN / 2019 has been registered in Clinical Trials Registration and such completion is included in the text. Thank you. It was corrected.

Line 284: change can’t for cannot.

Thank you. It was corrected.

Line 284-285: What about muscle spasticity after stroke? Most of the time patients with stroke have limitations in performing fingers extension because of the spasticity. Did you measure spasticity levels of the hand, and forearm before starting the study? I think that this is an important characteristic to take into consideration regarding improvements in muscle tone and joint mobility. If the authors did not measure spasticity, this should be added as a limitation of the study section. 

Yes, muscle tone was tested before the start of the research. We used a Modified Ashworth Scale (MAS) for this purpose. Thank you so much for this comment. I supplemented and corrected the necessary information in the subjects section.

Line 292-293: In addition to the dominant or non-dominant hand, the authors should specify the affected or non-affected side. Did the authors take this into consideration when assessing the patients? I mean, did you consider if the affected upper limb of the patients with stroke were their dominant or non-dominant side? How this could affect motor improvements of the upper limb? If not, this should be added in the limitations section too.

We considered whether the affected upper limb was dominant or non-dominant, but in fact, this is not clearly stated in the text. While correcting the manuscript, we introduced the necessary changes to the text and tables. The limitation of the study, however, was that only the affected upper limb was examined with regard to whether it was dominant or not. Examination of the opposite limb under conditions with and without stabilization would increase the observation effect/ strength of the study. The authors have completed the study limitation. Thank you so much for this comment.

Line 297: Correlated? The authors did not use any correlation test in the statistical analyses.

Indeed, no correlation tests were used for the statistical analysis. The use of the word correlation has been misused. I made necessary changes to the text.

Thank you very much for this suggestion.

Line 358: eliminate the or their.

Thank you very much. This suggestion was corrected.

Thank you very much for your time.

Reviewer 2 Report

This is an observational study on the correlation between muscle tone, trunk and "proximal" upper limb stabilization, and parameters of hand-wrist motor coordination, assessed instrumentally in a single acquisition, in sub-acute stroke patients. The results were compared to a control group of neurologically healthy subjects, showing that passive shoulder and trunk stabilization may lead a better hand and writ coordination in sub-acute stroke patients. The topic could be interesting, but in the way the manuscript is presented, it is hard to follow. The English style is of fair quality, although some language errors should be revised. However, the article globally doesn't follow scientific writing and the entire manuscript needs extensive revision. 

I was interested in your study, which I believe addresses an interesting subject. However, the text is difficult to follow which makes the paper unattractive to non-specialized people. Methods are presented in a confusing way, thus difficult in order to reproducibility.

Specific comments

The Title could be misleading. I would expect the study design in the title; population: subacute stroke patients.

The abstract should stand alone, but the study design and the methods are difficult to understand for me.

The abstract should be re-structured. The study aim is not the same as the one reported in the main text.

Line 29. In both groups? Please, specify it.

The introduction is short and contains a part of the methods.

Line 44 e 45: The concept of core stability is not related to this study topic. You'd better report state-of-art or studies focused on upper limb rehabilitation with novel approaches in subacute stroke subjects underlying what your work further offers to clinicians. (Goffredo et al. Kinematic parameters for tracking patient progress during upper limb robot-assisted rehabilitation: An observational study on subacute stroke subjects. Applied bionics and biomechanics. 2019 Oct 21;2019. Franceschini et al.  Upper limb robot-assisted rehabilitation versus physical therapy on subacute stroke patients: A follow-up study. Journal of bodywork and movement therapies. 2020 Jan 1;24(1):194-8.)

Line 61. ...discovered... by "her"; instead of "him"

Line 63. ..distal elbow joint... the term "distal" is not necessary.

Lines 73 to 92 should be reported in the Methods section.

Did the authors consider the age limit for the clinical/stroke group? If yes, please specify it.

I found somewhere through the text that authors considered a score limit of the Trunk control test and Fugle-Meyer Assessment score (I suppose the motor performance 0-66) for including the stroke subjects. A) This information is not included among the inclusion criteria; B) It is not clear in methods who performed the clinical assessments on the stroke subjects and when; C) The reference of the version of FMA applied in the study should be added; D) In Inclusion criteria should be reported also that only ischemic stroke was considered for this study.

In the Methods, Sub-sections should be labeled appropriately. A) Study design should be clearly described here; B) Ethical aspects should include also lines 134-136; C)Subjects should be labeled Statistical Analysis-Sample size calculation.

Figure 1. please refer to the CONSORT flow-chart as a valid format. It should be reported after paragraphs 119-123. However, a total of 100 subjects including both stroke and control ones were ONLY screened and NOT included in the study, and 66 patients were eligible and met the inclusion and exclusion criteria. The sample should not be described in the flow of participants.

Line 125. 6-7 weeks from stroke onset is not considered the CHRONIC stage of the disease. All your sample falls in the sub-acute stage.

Table 1. is presented in an unusual format. In the sample description of the clinical/stroke group, clinical characteristics must be presented in addition to demographic and biometric data.

Measurements:

Line 143. Which kind of sensors? IMU ones? Please specify it.

Line 149. "ECG" maybe "EMG" electrodes?

Results are too much explained. Please, avoid repeating information detectable from tables or report only the most significant one.

Discussion: I had expected more focus on the results:  comparison between clinical group and control group.

You'd better use "proximal" and "distal" upper extremity OR upper limb in the whole text.

Lines 372-376. Please rephrase.

Line 380. Or "their" or "the" choose one.

References:

Ref. 7. Correct the report the DOI  only rather than the link.

Lines 433-434. Space should be eliminated.

Ref.16 and 17. Please correct.

Author Response

Manuscript ID: jcm-1173910

Title (previous): Hand function evaluation by HandTutor device and a manual dynamometer in stroke patients in a supported and unsupported position

Full title (after change): Influence of a stable and unstable position of the trunk and upper limb on selected parameters of the wrist and hand motor coordination, grip strength, and muscle tension, in post-stroke and healthy subjects.

Dear Reviewers,

   Thank you very much for the analysis of our manuscript. We really appreciate your comments and indication of fragments that should be corrected and explained. Considering your suggestions, all mistakes were corrected. The introduction of corrections and changes in the text caused the numbering of the lines to shift. In response to reviewers' comments, it provides the original numbering. In order to avoid misunderstandings, changes introduced in the text are marked in blue and additionally, the manuscript was sent in the change tracking mode.

Reviewer #2:

Thank you very much for the very quick and thorough analysis of our manuscript.

The following comments and answers:

The Title could be misleading. I would expect the study design in the title; population: subacute stroke patients.

The title has been changed. The title seems to be more related to the content of the manuscript as it stands. Thank you very much for the suggestion.

The abstract should stand alone, but the study design and the methods are difficult to understand for me.

In the revised version, the study design is presented in detail. Like the methods that have been tweaked, supplemented, and hopes to make this project understandable. I used all the suggestions in the revised version of the work. Thank you very much.

The abstract should be re-structured. The study aim is not the same as the one reported in the main text.

The executive summary has been rebuilt. The purpose was corrected, and now it is consistent with that in the manuscript introduction. Thank you very much for this suggestion.

Line 29. In both groups? Please, specify it.

Thank you very much. This suggestion was corrected.

The introduction is short and contains a part of the methods.

All items that should be included in the Methods section have been removed from the introduction section. Thank you very much. This suggestion was corrected.

Line 44 e 45: The concept of core stability is not related to this study topic. You'd better report state-of-art or studies focused on upper limb rehabilitation with novel approaches in subacute stroke subjects underlying what your work further offers to clinicians. (Goffredo et al. Kinematic parameters for tracking patient progress during upper limb robot-assisted rehabilitation: An observational study on subacute stroke subjects. Applied bionics and biomechanics. 2019 Oct 21;2019. Franceschini et al.  Upper limb robot-assisted rehabilitation versus physical therapy on subacute stroke patients: A follow-up study. Journal of bodywork and movement therapies. 2020 Jan 1;24(1):194-8.)

I corrected and completed the introduction. Due to the fact that the research assumed an independent variable in the form of changes in position and included elements of body stabilization, I tried to explain the importance of stabilization for movement. Therefore, new references have appeared in the reference section.

Thank you for the proposition of articles, but they seem not quite appropriate to me to use the proposed references, especially since the proposed articles analyze the proximal and not the distal parts of the upper limb, which are the essence of our work. Instead, other references were used :

(41. Cotoros D. Biomechanical Analyzes of Human Body Stability and Equilibrium, Proceedings of the World Congress on Engineering 2010, Vol II, WCE 2010, June 30 - July 2, 2010, London, U.K.

  1. Knudson D. Fundamentals of Biomechanics, 2003 Kluwer Academics/Plenum Publishers, New York, ISBN 0-306-47474-3, pp.178-188.
  2. Farjoun N, Mayston M, Florencio LL, Fernández-De-Las-Peñas C, Palacios-Ceña D. Essence of the Bobath concept in the treatment of children with cerebral palsy. A qualitative study of the experience of Spanish therapists. Physiother Theory Pract. 2020 Feb 11;1-13. DOI: 10.1080/09593985.2020.1725943.
  3. Lennon S, Ashburn A. The Bobath concept in stroke rehabilitation: a focus group study of the experienced physiotherapists' perspective. Disabil Rehabil. 2000 Oct 15;22(15):665-74. DOI: 10.1080/096382800445461.
  4. Lennon S, Baxter D, Ashburn A. Physiotherapy based on the Bobath concept in stroke rehabilitation: a survey within the UK. Disabil Rehabil. 2001 Apr 15;23(6):254-62. DOI: 10.1080/096382801750110892.
  5. Luke C, Dodd KJ, Brock K. Outcomes of the Bobath concept on upper limb recovery following stroke. Clin Rehabil. 2004 Dec;18(8):888-98. doi: 10.1191/0269215504cr793oa.
  6. Graham JV, Eustace C, Brock K, Swain E, Irwin-Carruthers S. The Bobath concept in contemporary clinical practice. Top Stroke Rehabil. 2009 Jan-Feb;16(1):57-68. DOI: 10.1310/tsr1601-57.)

Thank you very much for this comment.

Line 61. ...discovered... by "her"; instead of "him"

“Brunnstrom in her 1970 book recalls the Souque's phenomenon, discovered by him in 1916 and consisting in that elevation of the affected arm frequently caused the paralyzed finger to extend. 18”

The phenomenon mentioned in Brunnstrum’s book was discovered by Souques in 1916, so the quoted passage should probably not be by her but by him.

Thank you for this review.

Line 63. ..distal elbow joint... the term "distal" is not necessary.

Thank you very much. This suggestion was corrected.

Lines 73 to 92 should be reported in the Methods section.

Thank you very much for this review. The inclusion criteria have been corrected, supplemented, and moved to the Methods section.

Did the authors consider the age limit for the clinical/stroke group? If yes, please specify it.

The age limit for the study groups is given in the inclusion criteria. Thank you very much. This suggestion was corrected.

I found somewhere through the text that authors considered a score limit of the Trunk control test and Fugle-Meyer Assessment score (I suppose the motor performance 0-66) for including the stroke subjects. A) This information is not included among the inclusion criteria; B) It is not clear in methods who performed the clinical assessments on the stroke subjects and when; C) The reference of the version of FMA applied in the study should be added; D) In Inclusion criteria should be reported also that only ischemic stroke was considered for this study.

  1. A) the inclusion criteria were supplemented with functional tests used to assess the functional status of patients and the required scores. B) in the Method section, I also added who and when conducted the clinical evaluation of patients; C) I added a reference to the FMA version; D) the inclusion criteria were supplemented with information about the type of stroke being studied.

Thank you very much for the suggestions. They have all been included.

In the Methods, Sub-sections should be labeled appropriately. A) Study design should be clearly described here; B) Ethical aspects should include also lines 134-136; C)Subjects should be labeled Statistical Analysis-Sample size calculation.

I made all appropriate corrections in the Methods section. A) the study design was described in detail; B) lines 134-136 were brought into Ethical aspects; C) the Subject section has been completed, corrected.

Thank you very much. All suggestions were corrected.

Figure 1. please refer to the CONSORT flow-chart as a valid format. It should be reported after paragraphs 119-123. However, a total of 100 subjects including both stroke and control ones were ONLY screened and NOT included in the study, and 66 patients were eligible and met the inclusion and exclusion criteria. The sample should not be described in the flow of participants.

As suggested, I used the CONSORT flowchart and placed it after points 119-123. If there is any inconsistency, it results from the broken lines during the applied corrections. I am not sure if this scheme is necessary at all in this case? Did I understand the suggestion? Thank you very much.

Line 125. 6-7 weeks from stroke onset is not considered the CHRONIC stage of the disease. All your sample falls in the sub-acute stage.

Thank you very much for this information. This suggestion was corrected.

Table 1. is presented in an unusual format. In the sample description of the clinical/stroke group, clinical characteristics must be presented in addition to demographic and biometric data.

The table has been corrected, moved, data on clinical characteristics supplemented in the text of the subjects section.

Thank you very much. This suggestion was corrected.

Measurements:

Line 143. Which kind of sensors? IMU ones? Please specify it.

If I understood the suggestion correctly, it was about the type of sensors used in the HandTutor glove. Information completed in the text, in the new "Apparatus" section (glove with sensitive electro-optical position, speed wrist, and finger movement sensors). Thank you very much. This part was corrected according to your suggestion.

Line 149. "ECG" maybe "EMG" electrodes?

Thank you very much. This suggestion was corrected.

Results are too much explained. Please, avoid repeating information detectable from tables or report only the most significant one.

Corrections in the results section were made according to your suggestions.

Discussion: I had expected more focus on the results:  comparison between clinical group and control group.

I have made additional comments in the discussion section. Thank you very much.

This suggestion was corrected.

You'd better use "proximal" and "distal" upper extremity OR upper limb in the whole text.

Thank you very much. This suggestion was corrected.

Lines 372-376. Please rephrase.

I edited. Hope the text quality is better.

Thank you very much for this comment.

Line 380. Or "their" or "the" choose one.

Thank you very much. This suggestion was corrected.

References:

Ref. 7. Correct the report the DOI  only rather than the link.

Thank you. It was corrected.

Lines 433-434. Space should be eliminated.

Thank you. It was corrected.

Ref.16 and 17. Please correct.

Thank you. It was corrected.

Thank you very much for your time.

Reviewer 3 Report

Overall Impression

While the topic may seems worthy of investigation the current manuscript has several red flags and it is very poorly conducted. I understand that the authors may have put a lot of effort to collect all these data but the manuscript is lacking in all sections. Below are some major and minor points to consider.

Major points

Abstract

Objective: Your objective is not clear. Your objective should be so clear that the reader should be able to guess what type of analysis will be performed. What is the primary objective and what is the primary outcome of interest?range of motion, hand dexterity, grip strength?

Design: Is this a measurement study? If yes, then you should follow the COSMIN checklist guidelines for performance-based tests. Again it is not clear how the authors had stroke patients and healthy patients as controls. You cannot match or compare healthy individuals and patients with stroke.

Results: As I said earlier your objective is not clear so it is very questionable what the p values represent here. I am afraid that your analysis is very shallow and the analysis needs to go beyond p values.

Conclusions: Your conclusions are clearly not supported by your data analysis.

Introduction: Your introduction does not serve it’s purpose and there are sections that do not belong in the introduction (e.g. inclusion criteria). The group inclusion criteria should be moved to the methods. 

Methods: 

Sample Size calculation: Your sample size calculation is clearly wrong. This is not an efficacy study but a measurement study. So your sample size calculation should be matching what is the primary objective of the study based on the design of the study. From the objective of the study it is very difficult to understand that because it is not clear and specific.

Flow chart/exclusion criteria/Line 120: What is functional condition?and why patients or subjects were excluded because of functional condition?

Line 124-128: This section belongs to the results not to the methods. The manuscript needs to follow reporting guidelines for observational/cross-sectional or measurement studies. The reporting quality of your manuscript is very poor.

Measurements: For each measurement you need to present how the test was administer, how was measured, referenced the measurement properties of each test (e.g Hand grip strength PMID: 31730754). It is not clear what was measured, finger and wrist ROM, hand dexterity, hand function as grip strength? 

Line 144: Why speed of movement and not quality of movement? Did the authors perform a hand dexterity test?

Statistical analysis: Why the 2 groups were compared? This is arguably very problematic comparing healthy and stroke individuals. What is the rationale behind that?

Results: The authors are not consistent with their terminology and they are always introducing new words. How coordination was measured? This was not mentioned in the objective or in the methods.

Table 2: Why the numbers are in red font? This is very simple descriptive statistics which does not indicate anything and no inferences can be made.

Line 261-262/Discussion: Your statement is not supported by your data analysis. This statement cannot be supported by a simple by a Mann-Whitney or sign rank test.  This statement will be supported by a regression analysis.

Minor points:

Please correct typos e.g line 30

Author Response

Manuscript ID: jcm-1173910

Title (previous): Hand function evaluation by HandTutor device and a manual dynamometer in stroke patients in supported and unsupported position

Full title (after change): Influence of a stable and unstable position of the trunk and upper limb on selected parameters of wrist and hand motor coordination, grip strength and muscle tension, in post-stroke and healthy subjects.

Dear Reviewers,

   Thank you very much for the analysis of our manuscript. We really appreciate your comments and indication of fragments that should be corrected and explained. Considering your suggestions, all mistakes were corrected. The introduction of corrections and changes in the text caused the numbering of the lines to shift. In response to reviewers' comments, it provides the original numbering. In order to avoid misunderstandings, changes introduced in the text are marked in blue and additionally, the manuscript was sent in the change tracking mode.

Reviewer #3:

Thank you very much for the very quick and thorough analysis of our manuscript.

The following comments and answers:

Abstract

Objective: Your objective is not clear. Your objective should be so clear that the reader should be able to guess what type of analysis will be performed. What is the primary objective and what is the primary outcome of interest? range of motion, hand dexterity, grip strength?

The aim of the work has been corrected, both in the abstract and in the introduction.

Thank you very much for this suggestion.

Design: Is this a measurement study? If yes, then you should follow the COSMIN checklist guidelines for performance-based tests. Again it is not clear how the authors had stroke patients and healthy patients as controls. You cannot match or compare healthy individuals and patients with stroke.

Indeed, the study design had not been clearly explained before. I made the appropriate corrections as below:

“This is an observational study. Measurements made in two different positions of the trunk and affected upper limb. It has been checked how the intervention (effectiveness of stabilization) translates into a change in parameters. The passive and active range of motion, the maximum range of movement (ROM), and the frequency or speed of movement at the wrist and fingers, as well as the grip force and the tension of the muscles deeply stabilizing the trunk and shoulder joint (dependent variables) were assessed among post-stroke patients in the sitting (non-stabilized) and supine (stabilized) position (independent variables). A group of neurologically healthy people was examined to assess whether neurological deficits in people after a stroke might affect the results of motor coordination and grip strength.”

Very often in clinical trials, there are people from the healthy group as a control sample and some patients. Healthy people were selected (age, BMI matched) for the control group to check whether the stable position of the trunk and the affected upper limb of people after a stroke significantly improves the parameters of motor coordination and grip force, and at the same time that the analysis of muscle tension will bring the researchers closer to understanding the essence of functional treatment of patients with impaired movement coordination. Thank you very much for this review.

Results: As I said earlier your objective is not clear so it is very questionable what the p values represent here. I am afraid that your analysis is very shallow and the analysis needs to go beyond p values.

I hope that explaining the study design and improving the purpose of the study justifies the value of probability. Thank you very much for this comment.

Conclusions: Your conclusions are clearly not supported by your data analysis.

I hope that explaining the study design and results can support the conclusions drawn.

Thank you very much for the very thorough analysis.

Introduction: Your introduction does not serve it’s purpose and there are sections that do not belong in the introduction (e.g. inclusion criteria). The group inclusion criteria should be moved to the methods.

The inclusion and exclusion criteria have been corrected, supplemented, and moved to the Methods section. Thank you very much for this review.

Methods:

Sample Size calculation: Your sample size calculation is clearly wrong. This is not an efficacy study but a measurement study. So your sample size calculation should be matching what is the primary objective of the study based on the design of the study. From the objective of the study, it is very difficult to understand that because it is not clear and specific.

This is an observational study. The sample size was calculated after the test was performed. A non-parametric test was used in the analyzes due to the skewed results of the measurements. On this basis, the sample size for which the power would be at a satisfactory level was estimated.

Thank you very much for this comment.

Flow chart/exclusion criteria/Line 120: What is a functional condition? and why patients or subjects were excluded because of functional condition?

In order to check selected movement parameters, such as the range of motion or the frequency and strength of the handgrip in a stable and unstable position of the body, a certain level of stiffness in the torso, and the ability to perform a movement task in the affected upper limb were needed. Functional capabilities were the inclusion criteria for the study.

Thank you very much for the very thorough analysis.

Line 124-128: This section belongs to the results not to the methods. The manuscript needs to follow reporting guidelines for observational/cross-sectional or measurement studies. The reporting quality of your manuscript is very poor.

Thank you very much. All suggestions were corrected.

Measurements: For each measurement you need to present how the test was administer, how was measured, referenced the measurement properties of each test (e.g Hand grip strength PMID: 31730754). It is not clear what was measured, finger and wrist ROM, hand dexterity, hand function as grip strength?

Thank you very much. All suggestions were corrected.

Line 144: Why the speed of movement and not the quality of movement? Did the authors perform a hand dexterity test?

I change the speed on the frequency of movement. The frequency is measured by the Hand Tutor device and the word speed can be abuse. Thank you so much for bringing this to my attention.

Statistical analysis: Why the 2 groups were compared? This is arguably very problematic comparing healthy and stroke individuals. What is the rationale behind that?

Healthy people were selected for the control group to check whether the stable position of the trunk and the affected upper limb of people after a stroke significantly improves the parameters of motor coordination, grip force, and muscle tension. Very often in clinical trials, there are people from the healthy group as a control sample and patients. This can bring the researchers closer to understanding the essence of functional treatment of patients with impaired movement coordination. Thank you very much for this comment.

Results: The authors are not consistent with their terminology and they are always introducing new words. How coordination was measured? This was not mentioned in the objective or in the methods.

Manuscript revision includes the required changes. Both the purpose and methods have been improved. Thank you very much for this comment.

Table 2: Why the numbers are in the red font? This is very simple descriptive statistics that do not indicate anything and no inferences can be made.

I removed the red font from the tables.

Thank you. The suggestions were corrected.

Line 261-262/Discussion: Your statement is not supported by your data analysis. This statement cannot be supported by a simple Mann-Whitney or sign rank test.  This statement will be supported by regression analysis.

The design of the analyzes does not allow the use of regression analysis and drawing conclusions based on it. The analysis is based on the comparison of groups and checking the stabilization effects. This explains why the analyzes based on the Mann Whitney and Wilcoxon tests were carried out. I made appropriate corrections in the results and discussion section.

In these circumstances, the analysis performed seems to be sufficient to draw conclusions. The topic requires further careful research. Thank you for the criticism and the opportunity to expand my knowledge.

Minor points:

Please correct typos e.g line 30

I reviewed the work and corrected typos and punctuation marks. If I have skipped it, I apologize, and please indicate it.

Thank you very much for your time.

Round 2

Reviewer 1 Report

The authors accomplished all my concerns. 

Author Response

Manuscript ID: jcm-1173910

Full title (after change): Influence of a stable and unstable position of the trunk and upper limb on selected parameters of wrist and hand motor coordination, grip strength and muscle tension, in post-stroke and healthy subjects.

Dear Reviewers,

   Thank you very much for the analysis of our manuscript.

Reviewer #1:

Thank you very much for accepting the manuscript correction.

Thank you very much for your time.

Reviewer 2 Report

Reference numbering among the whole text is uncorrected.

The introduction can be improved by presenting novel studies in litterature and giving more information about the novelty and the hypothesis of the study.

Was the back pain considered as the inclusion or exclusion criteria in the healthy group?

Table 1 should be presented in a standard format.

When authors talk about specific parameters this should indicated. Which kind of parameters? Biomechanical or Kinematic parameters? Muscle activity?

The type of sensors should be specified as requested in the previous revision. You would refer to the tool manual or contact the company for this detail.

Author Response

Manuscript ID: jcm-1173910

Full title (after change): Influence of a stable and unstable position of the trunk and upper limb on selected parameters of the wrist and hand motor coordination, grip strength, and muscle tension, in post-stroke and healthy subjects.

Dear Reviewers,

   Thank you very much for the analysis of our manuscript.

We really appreciate your comments and indication of fragments that should be corrected and explained. Considering your suggestions, all mistakes were corrected. Changes introduced in the text are marked in green and additionally, the manuscript was sent in the change tracking mode.

Reviewer #2:

Thank you very much for the very thorough analysis of our manuscript.

The following comments and answers:

Comments and Suggestions for Authors

Reference numbering among the whole text is uncorrected.

Reference numbers have been corrected throughout the work, placed in square brackets. In addition, I improved the order of references in the text and on the list of references.

Thank you very much for this comment.

The introduction can be improved by presenting novel studies in literature and giving more information about the novelty and the hypothesis of the study.

I have analyzed and completed the introduction based on the suggestion that it would be good to write about new working methods and better support my own hypothesis. There are new references in the reference part:

  1. Weiler J.; Gribble P.L.; Pruszynski J.A. Rapid feedback responses are flexibly coordinated across arm muscles to support goal-directed reaching. J Neurophysiol. 2018; 119(2):537-547.
  2. Rand M.K.; Rentsch S. Eye-Hand Coordination during Visuomotor Adaptation with Different Rotation Angles: Effects of Terminal Visual Feedback. PLoS One. 2016; 11(11):e0164602.
  3. Delph M.A.;, Fischer S.A.; Gauthier P.W.; et al. A soft robotic exomusculature glove with integrated sEMG sensing for hand rehabilitation. IEEE Int Conf Rehabil Robot. 2013 Jun;2013: 6650426.
  4. Golaszewski S.; Kremser C.; Wagner M.; et al. Functional magnetic resonance imaging of the human motor cortex before and after whole-hand afferent electrical stimulation. Scandinavian Journal of rehabilitation medicine 1999; Vol. 31, Iss. 3: 165-173.
  5. Oluigbo C. O.; Salma A.; Rezai A.R. Deep brain stimulation for neurological disorders. IEEE reviews in biomedical engineering 2012; Vol. 5: 88-99.
  6. Katayama Y. Deep brain stimulation therapy for involuntary movements. Rinsho shinkeigaku = Clinical neurology. 2001; Vol. 41, Iss. 12: 1079-1080.

They show the possibilities of working to improve the function of the hand, but at the same time, our work is not a duplication or repetition similar to the presented works.

Thank you very much for this suggestion.

Was the back pain considered as the inclusion or exclusion criteria in the healthy group?

I rather forgot to include low back pain as a factor that excludes from the study, not only healthy people but also stroke patients.

It was a mistake and was corrected. Thank you.

Table 1 should be presented in a standard format.

Table 1 was corrected. Thank you for this comment.

When authors talk about specific parameters this should indicated. Which kind of parameters? Biomechanical or Kinematic parameters? Muscle activity?

We have added the following information:

Line 182: “A Hand Tutor was used to measure the kinematic parameters like the range of passive and active movement, deficits of movement (sensitivity: 0,05 [mm] of wrist and fingers Ext./Flex) as well as the speed/frequency of movement (motion capture speed: up to 1 [m/sec]).”

Thank you very much for this comment.

The type of sensors should be specified as requested in the previous revision. You would refer to the tool manual or contact the company for this detail.

I used the HandTutor tool manual to complete the data.

“A Hand Tutor device composed of a safe and comfortable glove (with sensitive electro-optical sensors evaluating a position, speed wrist, and finger movement; power supply: voltage: 5[V] DC, rated current input: 300[mA]),….. “

Thank you very much for this suggestion.

Thank you very much for your comments and kind words. We hope that the current form of the manuscript has met your expectations. Thank you very much for your time.

Reviewer 3 Report

I understand that the authors put a tremendous effort to collect the current data however, my comments were not addressed or the author do not fully understand what was requested.

Author Response

Manuscript ID: jcm-1173910

Full title (after change): Influence of a stable and unstable position of the trunk and upper limb on selected parameters of the wrist and hand motor coordination, grip strength, and muscle tension, in post-stroke and healthy subjects.

Dear Reviewers,

   Thank you very much for the analysis of our manuscript. We really appreciate your comments and indication of fragments that should be corrected and explained. Changes introduced in the text are marked in green and additionally, the manuscript was sent in the change tracking mode.

Reviewer #3:

Comments and Suggestions for Authors

I understand that the authors put a tremendous effort to collect the current data however, my comments were not addressed or the author do not fully understand what was requested.

Thank you very much for the very thorough analysis of our manuscript.

I would like to explain the changes made in line with your suggestions, once again.

    The original version of the work under the title: "Hand function evaluation by HandTutor device and a manual dynamometer in stroke patients in supported and unsupported position" has been corrected.

    Starting from the title change, which now reads: "Influence of a stable and unstable position of the trunk and upper limb on selected parameters of the wrist and hand motor coordination, grip strength and muscle tension, in post-stroke and healthy subjects" and to standardize the purpose of the research contained in the summary and the introduction, the description of the study design was also improved.

As previously indicated: „Your objective is not clear. Your objective should be so clear that the reader should be able to guess what type of analysis will be performed. What is the primary objective and what is the primary outcome of interest? range of motion, hand dexterity, grip strength?

In the current version, the goal has been formulated as follows:

” The aim of the study was to evaluate the influence of the stable position of the trunk and affected upper limb (dominant or non-dominant) on the parameters of hand and wrist movement coordination, grip strength, and muscle tension in subacute post-stroke patients compared to healthy subjects.”

This version informs why to whom and how the analysis will apply. Additionally, he emphasizes that this will be the evaluation of parameters after introducing a change, which is a stable position of the examined person.

Sample Size calculation: Your sample size calculation is clearly wrong. This is not an efficacy study but a measurement study. So your sample size calculation should be matching what is the primary objective of the study based on the design of the study. From the objective of the study, it is very difficult to understand that because it is not clear and specific.

This is not a measurement study but an observational study. Sample size can be counted both before and after testing. The sample size was calculated after the test was performed. A non-parametric test was used in the analyzes due to the skewed results of the measurements. On this basis, the sample size for which the power would be at a satisfactory level was estimated. Our groups are larger than estimated.

Study design

    Indeed, the study design had not been clearly explained before. I have made corrections in my work as described below:

“This is an observational study. Measurements made in two different positions of the trunk and affected upper limb. It has been checked how the intervention (effectiveness of stabilization) translates into a change in parameters. The passive and active range of motion, the maximum range of movement (ROM), and the frequency of movement at the wrist and fingers, as well as the grip force and the tension of the muscles deeply stabilizing the trunk and shoulder joint [dependent variables] were assessed among post-stroke patients in the sitting (non-stabilized) and supine (stabilized) position [independent variables]. A group of neurologically healthy people was examined to assess whether neurological deficits in people after a stroke might affect the results of motor coordination and grip strength.”

     Very often, clinical trials compare clinical cases with healthy controls. Healthy people were selected (age, BMI matched) for the control group to check whether the stable position of the trunk and the affected upper limb of people after a stroke significantly improves the parameters of motor coordination and grip force, and at the same time that the analysis of muscle tension will bring the researchers closer to understanding the essence of functional treatment of patients with impaired movement coordination.

As previously indicated:

“Your statement is not supported by your data analysis. This statement cannot be supported by a simple Mann-Whitney or sign rank test.  This statement will be supported by regression analysis.”    I explain:

The design of the analyzes does not allow the use of regression analysis and drawing conclusions based on it. The analysis is based on the comparison of groups and checking the stabilization effects. This explains why the analyzes based on the Mann Whitney and Wilcoxon tests were carried out. I made appropriate corrections in the results and discussion section. In these circumstances, the analysis performed seems to be sufficient to draw conclusions. The topic requires further careful research. Thank you for the opportunity to expand my knowledge.

Thank you for the very good words you directed to our tremendous effort. We corrected everything we understood. Nevertheless, we hope that the current form of the manuscript has met your expectations. If some comments did not meet your expectation please let us know so we can improve our manuscript according to your suggestions.